# Effects of 3D Postural Correction and Abdominal Muscle Contraction on the Symmetry of the Transverse Abdominis and Spinal Alignment in Patients with Idiopathic Scoliosis

**DOI:** 10.3390/ijerph20065016

**Published:** 2023-03-12

**Authors:** Sung-Young Yoon, Sang-Yeol Lee

**Affiliations:** 1Department of Physical Therapy, Busan Health University, Busan 49318, Republic of Korea; yoonsypt@bhu.ac.kr; 2Department of Physical Therapy, Graduated School of Kyungsung University, Busan 48434, Republic of Korea; 3Department of Physical Therapy, Kyungsung University, Busan 48434, Republic of Korea

**Keywords:** scoliosis, 3D postural correction, ultrasound, transversus abdominis, symmetry

## Abstract

This study aimed to investigate the effectiveness of 3D postural correction (3DPC) using corrective cushions (CCs) and abdominal muscle contraction (AMC) on the thickness symmetry of the transversus abdominis (TrA) and spinal alignment in patients with idiopathic scoliosis (IS). In the first experiment, ultrasound measurements were taken of the TrA thickness on both the convex and concave sides of the lumbar curve in the supine position during AMC and non-AMC without 3DPC, and during AMC and non-AMC with 3DPC using CCs, in 11 IS patients. In the second experiment, 37 IS patients participated in a four-week 3DPC exercise program that aimed to maintain TrA thickness symmetry based on the results of the first experiment. The study found that TrA thickness symmetry significantly increased after 3DPC using CCs and combined with AMC (*p* < 0.05). Additionally, the Cobb angles and trunk rotation angles showed significant decreases, and trunk expansion showed a significant increase (*p* < 0.05). These results indicate that the simultaneous application of 3DPC and AMC is the most effective way to achieve TrA thickness symmetry in IS patients. Therefore, 3DPC and AMC should be considered as crucial elements in exercise interventions for IS patients.

## 1. Introduction

Idiopathic scoliosis (IS) refers to three-dimensional (3D) deformities of the trunk and spine resulting from various causes that occur during growth and throughout life. It is characterized not only by deformities of the lateral spinal curve and deviations from the anatomical central axis in the frontal plane but also by deformities of the vertebral body rotation and loss of the normal curve state in the sagittal plane [1,2,3].

IS can cause secondary problems due to abnormal structural changes in the vertebral body and disks, and can gradually lead to changes in trunk interactions. As a result, it may increase trunk instability, thereby causing asymmetries in shoulder height, ribs, trunk shape, and musculoskeletal system [4,5].

IS can affect the posture of adolescents, promoting a negative self-image and disturbing self-esteem, which could potentially lead to psychological problems [6,7,8]. If IS is not treated properly, it may lead to serious deformities of the trunk and reduced chest circumference and motor ability. It can affect quality-of-life aspects such as general health and work capabilities [9,10]. Other studies have found that trunk asymmetries (e.g., thickness, power) and weakness or improper function (e.g., onset timing, recruitment patterns) of abdominal muscles (oblique external (OE), oblique internal (OI), and transversus abdominis (TrA)) were the cause of spinal deformities [11,12,13,14,15].

Ultrasound imaging can easily and safely assess the thickness and location of, and changes to, deep abdominal muscles [16,17]. Linek et al. [18] compared the thickness of the abdominal muscles of adolescents with IS with that of a control group using ultrasound, and found that the OE, OI, and TrA muscles were overall thinner bilaterally in the IS group. Other studies have reported differences in thickness between the left and right sides of the TrA when the abdominal muscles of IS patients are contracted [16,19]. Since the difference in thickness of the TrA leads to the asymmetry of trunk and spine, treatment should focus on exercises targeting symmetry.

To treat IS, surgical intervention, physiotherapeutic scoliosis-specific exercise (PSSE), and conservative interventions, such as braces, can be used [20,21,22]. The purpose of conservative interventions is to prevent not only curve progression, which is the primary aim, but also potential complications such as postural asymmetry, changes in trunk appearance, back pain, and adverse psychosocial effects [9,23,24,25]. Conservative intervention methods include a PSSE specifically designed for IS [26,27,28,29]. The common principles of PSSE are 3D self-correction, patient education, training in activities of daily living, and stabilization of the corrected posture [22,29,30]. The most crucial element of these common principles is 3D self-correction, which refers to correction along with autonomous muscle contraction using external aids, such as corrective cushions (CCs). Several studies have emphasized the importance of 3D self-correction and symmetric muscle contraction, which constitute the basic approach of conservative IS treatments [19,31,32].

Although some previous studies have verified the thickness asymmetry between the left and right sides of the OE, OI, and TrA muscles of IS patients, few studies have investigated the symmetry of abdominal muscle thickness on the concave and convex sides of the lumbar curve when three-dimensionally aligned in real-time. We hypothesized that better TrA symmetry was achieved through 3D postural correction (3DPC) and AMC, and that this may have the beneficial effect of decreasing curve severity. Thus, this study aimed to quantitatively investigate TrA thickness symmetry when IS patients were 3D-aligned. Furthermore, the study aimed to assess the effectiveness of a 3D postural correction (3DPC) intervention in maintaining symmetry.

## 2. Materials and Methods

### 2.1. Participants

This study involved patients living in Busan, South Korea, diagnosed with idiopathic scoliosis (IS) between the ages of 10 and 30 years. These patients had Cobb angles ranging from 10° to 45°, and their accompanying vertebral body rotation and lateral deviation were confirmed through radiological findings. The patients had no history of braces or other conservative treatments. All curve types were recruited according to the Rigo classification [29,33,34,35]. This classification method uses clinical and radiological criteria, and is designed to overcome the limitations of the classification method that focuses on surgical treatment and the promotion of conservative treatment methods. All the patients were fully informed of the experimental procedures and provided informed consent before participating. Patients with functional scoliosis due to muscle imbalance or pain, congenital scoliosis, or neuromuscular scoliosis, patients with recent neurological or orthopedic diseases of the upper extremities, lower extremities, or lower back, patients with a history of spinal surgery, and patients with differences in leg length were excluded from the study. Basic information on the patients, including age, height, weight, and body mass index (BMI), was collected. To determine the sample size, the G-Power program (Kiel University, Germany) was used [36]. For the first experiment, the sample size was calculated to compare the means between conditions, with an effect size of 0.9, a significance level of 0.05, a power of 80%, and a dropout rate of 10%. For the second experiment, the sample size was calculated to compare the intervention effect, with an effect size of 0.9, a significance level of 0.05, a power of 80%, and a dropout rate of 20%.

### 2.2. Design and Setting

Two experiments were conducted in this study. In the first experiment, ultrasonic equipment was used to measure the TrA thickness on the convex and concave sides of the lumbar curve of 11 IS patients for four conditions. The four conditions are abdominal muscle contraction (AMC) and non-AMC without 3DPC, and during AMC and non-AMC with 3DPC using CCs, while in the supine position. The Rigo classification method, which divides the scoliosis curves into Rigo type A, Rigo type B, Rigo type C, and Rigo Type E was used. Comparisons of TrA thickness symmetry between the experimental conditions were performed.

The posture of each patient was measured with the patient in the supine position with a 45° hip joint flexion and a 90° knee joint flexion while the feet were shoulder-width apart. The measurement was performed at the end of normal expiration during comfortable breathing, with the hands placed on the sides of the trunk. Prior to measurement, a pressure biofeedback unit (Stabilizer, Chattanooga Group Inc., Hixson, TN, USA) was used to maintain 40 mmHg while the TrA was contracted [37,38].

The four conditions (AMC/non-AMC without 3DPC and AMC/non-AMC with 3DPC) were randomly chosen during the measurements (Figure 1). Three measurements were taken for each condition, with three minutes of rest between them.

To measure the TrA thickness, the perpendicular distance of the TrA at a site 1.5 cm away from the muscle fascia junction was measured on captured images using Image J (National Institutes of Health, Bethesda, MD, USA) [17,19,39,40]. The TrA symmetry ratio was calculated as TrA thickness on the convex side of the lumbar curve/TrA thickness on the concave side of the lumbar curve (Figure 2).

In the second experiment, the Cobb angles, trunk rotation angles, and trunk expansion were measured using X-ray imaging to assess the effectiveness of 3DPC in maintaining TrA thickness symmetry based on the first experiment results (Figure 3). Thirty-seven IS patients attended a 3DPC exercise program consisting of warm-up, main, and cool-down exercises. One-hour sessions were held twice a week for four weeks in a rehabilitation center (Table 1).

In the warm-up exercise, self-correction was performed while keeping the pelvis stable through a pelvic correction after breathing education [41]. In the main exercise, load was gradually increased from the supine position to sitting and standing positions. A CC was used according to each curve pattern for 3D alignment in the supine position. The corrected posture was then maintained without CCs in sitting and standing positions, and a mirror was used for visual feedback. Self-elongation, deflection, and de-rotation were performed to align the lumbar curve with the sagittal, frontal, and transverse planes, and then symmetric expansion was induced in the front and back, left and right, and up and down directions while focusing on the concave side of the curve through rotational angular breathing. The corrected posture was maintained as much as possible using isometric tension by slowly breathing during expiration from the expanded posture made through inspiration [29]. The cool-down exercise phase lasted 10 min, during which the self-corrected posture was maintained while performing a task-oriented exercise, such as a sit-to-stand exercise or walking and standing on an unstable floor. All procedures were conducted with the oversight of a specially trained physiotherapist.

### 2.3. Equipment and Data Collection

#### 2.3.1. Ultrasound Device

To measure the TrA thickness, ultrasonic equipment (F31; Hitachi-Aloka Medical, Tokyo, Japan) with a 10 MHz linear probe was used in dual-scan mode. The probe was placed 2.5 cm in the anterior and medial directions from the midpoint between the 12th rib and the iliac crest in the transverse direction [42,43,44,45,46]. Image J software (National Institutes of Health, Bethesda, MD, USA) was used for thickness measurements on captured images.

#### 2.3.2. Cobb Angle

Cobb angles, which can objectively determine changes in the curvature of the spine, were measured using X-ray equipment (SIG-40-525; Ecoray Inc., Seoul, Korea). X-ray images were acquired at a medical institution in Busan, South Korea, while the patients were in a standing position. The patients were advised of necessary precautions during image acquisition. The Cobb angle between intersecting lines drawn perpendicular to the upper-end vertebral plate of the upper vertebra and the lower-end vertebral plate of the lower vertebra was measured [47,48,49,50,51].

#### 2.3.3. Trunk Rotation Angle

A Bunnell scoliometer (Orthopedic Systems, Inc., Haywood, CA, USA) was used to measure the trunk rotation angle caused by scoliosis. Bonagamba et al. [52] reported high inter- and intra-rater reliability of the scoliosis goniometer. A forward bending test was performed with the patients putting their feet together, and the recessed area of the scoliometer was located in the spinous process. The steepest slope at the thoracic and lumbar blocks was then measured [53].

#### 2.3.4. Trunk Expansion

For trunk expansion, the axilla, xiphoid process, and waist areas were measured [54,55]. The measurements were performed by placing a tape measure horizontally on the trunk. The average value of three measurements at the time of maximum inspiration was calculated. All measurements were performed by a single therapist with 10 years of clinical experience.

#### 2.3.5. Corrective Cushion

A CC, which is a passive correction tool, was employed for 3DPC in the supine position. Each CC was wedge-shaped and 12 cm wide, 12 cm long, and 6 cm high. It was used to perform de-rotation, as it was positioned only on one side of the thoracic and lumbar vertebrae and pelvis. Its depth was adjusted by inserting it until the trunk was horizontal in relation to the floor surface by inducing vertebral de-rotation and deviation. Its surface was made of a non-slip material, and its hardness was maintained to achieve symmetric alignment of the trunk [56].

### 2.4. Data Analysis

Statistical analysis was performed using IBM SPSS Statistics version 25.0 (IBM Corp., Armonk, NY, USA). Two-way analysis of variance (ANOVA) was performed to compare the symmetry of TrA thickness on the convex and concave sides of the lumbar curve during AMC/non-AMC without 3DPC and the TrA symmetry on the convex and concave sides of the lumbar curve during AMC/non-AMC with 3DPC using a CC. In order to compare the differences in muscle thickness symmetry for AMC presence or absence and 3DPC presence or absence, a paired *t*-test was conducted separately for each group. Cronbach’s alpha and intraclass correlation coefficients (ICC 3,1) were calculated to assess the reliability of the measurement method. Multiple measurements were performed to test reliability. A paired *t*-test was conducted to examine whether there was a significant difference in symmetry between the pre- and post-intervention measures. As the xiphoid process and waist trunk expansion data were not normally distributed, a Wilcoxon signed-rank test, which is a nonparametric test, was used. The level of statistical significance was set to α = 0.05.

### 2.5. Ethical Considerations

The study was conducted in accordance with the Declaration of Helsinki, and the Institutional Review Board of Kyungsung University approved the study (KSU-19-12-001, 21 January 2020).

## 3. Results

### 3.1. General Characteristics of the Patients

Of the 11 participants in the first experiment, two were male and nine were female. The curve types were Rigo type A in 3 patients, Rigo type B in 5 patients, Rigo type C in 1 patient, and Rigo type E in 2 patients. The general characteristics of the participants in the first experiment are shown in Table 2.

Of the 37 participants in the second experiment, 6 were male and 31 were female. The curve types were Rigo type A in 13 patients, Rigo type B in 12 patients, Rigo type C in 5 patients, and Rigo type E in 7 patients. Risser signs were recorded as follows: Risser sign 3 was observed in 5 patients, Risser sign 4 in 10 patients, and Risser sign 5 in 22 patients. The general characteristics of the participants in the second experiment are shown in Table 3.

### 3.2. Reliability of the Measurement Method (TrA Thickness and Cobb Angle)

To analyze the reliability of the TrA thickness measurement method, the same ultrasound image was measured three times. The results showed that the reliability coefficient (Cronbach’s alpha) was 0.989. The intra-class coefficient (ICC 3,1) was 0.986 (95% CI 0.951–0.997), when three examiners measured the thickness of randomly selected TrA.

To analyze the reliability of the Cobb angle measurement method, the same X-ray image was measured three times. The results showed that the reliability coefficient (Cronbach’s alpha) was 0.988. The intra-class coefficient (ICC 3,1) was 0.979 (95% CI 0.911–0.997), when three examiners measured the Cobb angle of randomly selected X-ray images.

### 3.3. Comparison of TrA Thickness Symmetry Accoring to 3DPC and AMC (First Experiment)

TrA thickness symmetry significantly increased after 3DPC using CCs and after AMC. There was no statistically significant interaction between 3DPC and AMC (Table 4) (Figure 4). There was a significant difference in TrA thickness symmetry according to 3DPC using CCs between the AMC and non-AMC conditions. There was also a significant difference in TrA thickness symmetry according to AMC between 3DPC using CCs and no 3DPC.

### 3.4. Comparison of Parameters before and after the 3DPC Exercise Program (Second Experiment)

The Cobb angles and trunk rotation angles showed statistically significant decreases, while the trunk expansion showed a statistically significant increase after the four-week 3DPC exercise program (Table 5).

## 4. Discussion

This study conducted two experiments to evaluate the symmetry of TrA thickness in IS patients according to 3DPC and AMC, and the effectiveness of a 3DPC exercise program in maintaining symmetry. TrA symmetry is important to provide spinal stability to IS patients with TrA asymmetry and to prevent secondary imbalance. TrA is one of the deep muscles that provide spinal stability, and if TrA is asymmetric, it causes imbalance in the spine and trunk [57].

The first experiment was conducted to assess the symmetry of TrA thickness on the concave and convex sides of the lumbar curve according to 3DPC with CCs and AMC using ultrasound imaging. In IS patients, the TrA was most asymmetrical at rest, both 3DPC and AMC separately increased TrA symmetry, and the greatest symmetry was achieved when combining 3DPC and AMC.

Some previous studies utilized ultrasound to examine changes in patients with scoliosis in response to interventions. Linek et al. [16] found changes in abdominal muscle symmetry in mild adolescent idiopathic scoliosis according to posture, and Linek et al. [18] found that patients with adolescent idiopathic scoliosis displayed higher activity of the OE, OI, and TrA muscles on the right side during the active straight-leg raise. However, our study differs from previous studies in that we investigated the immediate effects of 3DPC using CC and AMC on TrA symmetry in patients with IS using ultrasound. Our results showed that the simultaneous application of 3DPC using CC and AMC immediately resulted in the greatest improvement in TrA symmetry on both the concave and convex sides of the lumbar curves in patients with IS.

The TrA is one of the deep trunk muscles, which plays an important role in lumbar stabilization as it is attached to the thoracolumbar fascia and is contracted first to provide spinal stability [57,58,59,60,61,62]. TrA thickness asymmetry may result in reduced balance, postural control ability, and stability during exercise and increased postural sway, all of which are observed in IS patients [18,63,64]. Kim et al. [19] verified the asymmetries in TrA thickness in IS patients and suggested a therapeutic method that could maintain trunk symmetry to have a normal operation of the TrA and prevent structural changes in the spine. A therapeutic method is important for preventing a vicious cycle of TrA thickness asymmetry and secondary asymmetries in IS patients [65].

In this study, there was a significant difference in the symmetry of TrA thickness on the concave and convex sides of the lumbar curve when 3DPC using CCs was applied, which was the first experimental condition. This was because vertebral deviation and de-rotation aligned the vertebrae in the frontal and horizontal planes as well as making structural symmetry of the TrA origin and insertion when performing 3DPC, thereby affecting the symmetry of TrA thickness on the concave and convex sides of the lumbar curve.

Braces and specially designed physiotherapy exercises are used as conservative treatments of scoliosis. Braces employ biomechanical factors, such as deviation, three-point pressure, and pairs of forces [33,34,66]. The CCs used in this study are considered to function as pads that provide pressure and force, which is the same in the production principle of braces. Zheng et al. [67] and Weinstein et al. [68] suggested that interventions using both braces and exercises could contribute to the improvement of scoliosis. However, despite the advantages of braces in improving spinal deformities and cosmetic aspects, exercise interventions are more beneficial from functional and psychological viewpoints, such as quality-of-life improvements. The use of braces may limit voluntary movement and cause psychological burdens due to aesthetic issues. The correction exercise method used in this study can overcome the limitations of braces and may result in voluntary changes in patients who can recognize symmetry.

AMC, which was the second experimental condition in this study, promoted the symmetry of TrA thickness on the concave and convex sides of the lumbar curve. The contraction of the TrA, which belongs to the antigravity muscles, resulted in greater symmetry in the standing position than in the supine position, indicating that the TrA, like other antigravity muscles, is contracted in a standing position to maintain posture [16]. Although this study used the supine position, which does not cause the TrA to contract, symmetry was achieved through active contraction of the TrA. This is because the TrA employs a symmetric contraction strategy by forming a musculofascial corset when it is contracted to prevent lumbar spine torsion and achieve trunk stabilization [14,16,69,70]. This symmetric contraction maintained the corrected posture through an active contraction in 3D self-correction, which is in line with the principles of conservative treatment methods for scoliosis. In particular, although there was no interaction between 3DPC and AMC in the first experiment when both conditions were applied at the same time, the TrA contracted most symmetrically on the concave and convex sides of the lumbar curve. This indicated that if the contraction of the TrA, which maintained symmetry only through the active contraction, occurred after adjusting the direction of vertebrae using CCs, the position and length of the TrA were maintained, promoting easy symmetrical contraction. This suggests that 3DPC and TrA contraction should be incorporated into exercise interventions for scoliosis.

The second experiment in this study verified the effectiveness of a four-week 3DPC exercise program in maintaining TrA thickness symmetry. A comparison of radiological findings before and after the intervention confirmed that the Cobb angles and trunk rotation angles decreased, while a comparison of clinical findings showed that trunk expansion increased. In this experiment, curve patterns were classified using the Rigo classification, thereby increasing the objectivity of the experimental design, a lack of which has been a limitation of previous studies on scoliosis. The patterns were divided into Rigo type A, Rigo type B, Rigo type C, and Rigo type E. Three-dimensional alignment using CCs was performed, and symmetry, which was important in the first experiment, was applied according to each curve pattern. The first experiment verified the TrA thickness symmetry in real-time, while the second experiment resulted in clinical and radiological changes through an exercise program aimed at maintaining TrA thickness symmetry. In other words, the symmetric contraction of the TrA had an impact on the clinical and radiological changes and was confirmed as an important factor for improving scoliosis. Although the patients who participated in the second experiment showed various levels of skeletal maturity, improvement was shown among the patients. Recent studies support the findings of this experiment, indicating that conservative treatment is effective even when bone maturity is reached [71,72].

This study has certain limitations. Due to the small number of participants, the results cannot be generalized, and differences in effects according to each curve pattern could not be distinguished. Although the subjects of the first and second experiments were recruited differently, it is necessary to recruit the same subjects in further studies. Further research is needed to confirm the effects of the combination of 3DPC and AMC on correcting spinal alignment through comparison with a control group. Various exercise programs that could be applied to scoliosis patients could be tested using 3DPC in various postures. Moreover, the intervention period could be longer, and a post-program evaluation could be conducted to assess the continuity of the intervention effects.

## 5. Conclusions

This study found that the application of 3DPC using CCs and AMC at the same time resulted in the greatest symmetry of TrA thickness on the concave and convex sides of the lumbar curves of IS patients. TrA symmetry is important to provide spinal stability and to prevent secondary imbalance for IS patients. Moreover, the four-week 3DPC exercise program aimed at maintaining this symmetry resulted in radiologically and clinically confirmed improvements. These results suggest that 3DPC and AMC should be used as key elements of conservative interventions for IS patients. The results can be used as the clinical foundation for future IS treatment directions.

## Figures and Tables

**Figure 1 ijerph-20-05016-f001:**
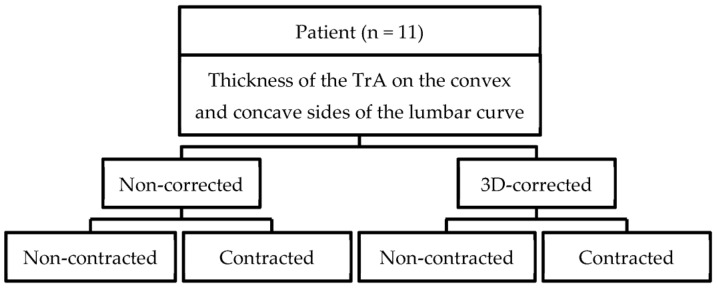
Diagram of the design of the first experiment.

**Figure 2 ijerph-20-05016-f002:**
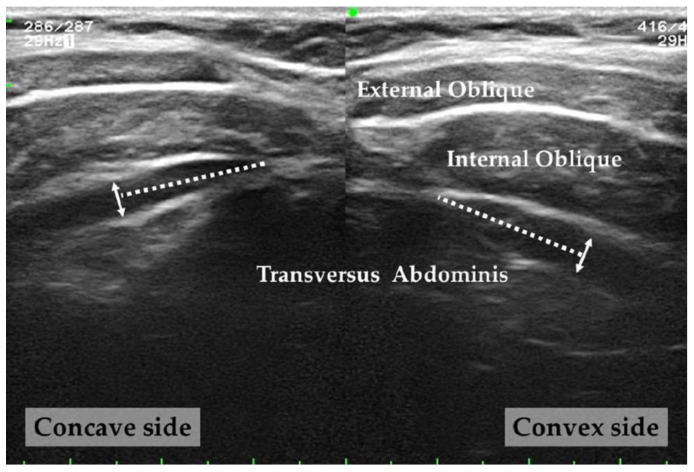
Ultrasound imaging measurement.

**Figure 3 ijerph-20-05016-f003:**
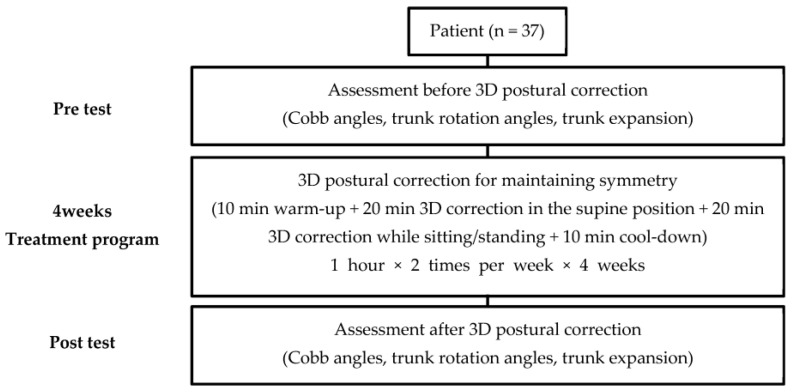
Diagram of the design of the second experiment.

**Figure 4 ijerph-20-05016-f004:**
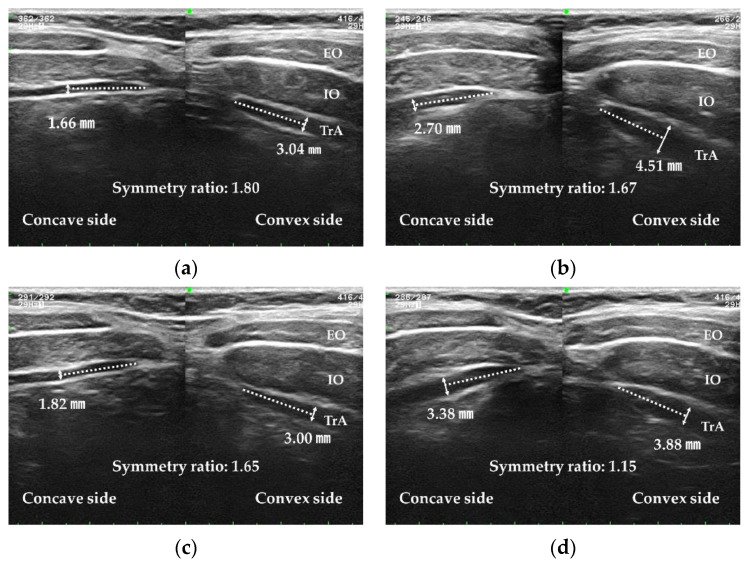
Results of an ultrasound imaging measurement. The symmetry ratio was calculated as the transversus abdominis thickness on the convex side of the lumbar curve divided by the thickness on the concave side. EO: external oblique; IO: internal oblique; TrA: transversus abdominis. (**a**) No correction and no contraction; (**b**) no correction and contraction; (**c**) correction and no contraction; and (**d**) correction and contraction.

**Table 1 ijerph-20-05016-t001:** General characteristics of the participants in the second experiment (*n* = 37).

Exercise	Exercise Program	Frequency
Warm-up (10 min)	Static stretching Breathing Pelvic self-corrections	10 s, 5 repetitions, 3 sets
Main exercise (40 min)	Semi-hanging Supine Sitting in a chair Standing	10 s, 10 repetitions, 3 sets
Cool-down (10 min)	Sit-to-stand Standing on an unstable floor Walking	30 s, 5 repetitions, 3 sets

**Table 2 ijerph-20-05016-t002:** General characteristics of the participants in the first experiment (*n* = 11).

Variable		Mean ± SD (Min–Max)
Age (years)		22.6 ± 4.4 (15–31)
Height (cm)		163.9 ± 6.4 (155–173)
Weight (kg)		55.2 ± 9.9 (39–74)
BMI (kg/m^2^)		20.4 ± 2.3 (15.43–24.73)
Cobb angle (°)	Thoracic	30.6 ± 12.1 (12–49)
	Lumbar	26.6 ± 10.7 (11–51)
Trunk rotation angle (°)	Thoracic	7.5 ± 3.7 (2–13)
	Lumbar	6.5 ± 3.7 (1–12)

All values are means ± standard deviations. BMI: body mass index.

**Table 3 ijerph-20-05016-t003:** General characteristics of the participants in the second experiment (*n* = 37).

Variable		Mean ± SD (Min-Max)
Age (years)		22.0 ± 6.6 (13–36)
Height (cm)		164.8 ± 6.5 (150–179)
Weight (kg)		51.1 ± 8.4 (33–80)
BMI (kg/m^2^)		19.1 ± 2.2 (14.67–25.32)
Cobb angle (°)	Thoracic	33.3 ± 12.2 (12–56)
	Lumbar	28.3 ± 11.0 (10–63)
Trunk rotation angle (°)	Thoracic	8.0 ± 4.2 (1–18)
	Lumbar	6.8 ± 4.2 (1–17)
Trunk expansion at rest (cm)	Axilla	79.0 ± 5.7 (68–94.5)
	Xiphoid process	69.8 ± 6.6 (61–91)
	Waist	65.5 ± 6.2 (56–82.5)

All values are means ± standard deviations. BMI: body mass index.

**Table 4 ijerph-20-05016-t004:** Two-way ANOVA results of the interaction effects between 3D postural correction and abdominal muscle contraction (*n* = 11).

	No Contraction	Contraction	*p*
No correction	1.42 ± 0.20	1.24 ± 0.12	<0.01 *
3D correction	1.30 ± 0.15	1.12 ± 0.09	<0.01 *
*p*	<0.01 *	<0.01 *	

All values (except *p*-values) are means ± standard deviations. * Significant at the 5% level. Transverse abdominis symmetry ratio: the closer the quotient to 1, the greater the TrA thickness symmetry.

**Table 5 ijerph-20-05016-t005:** Comparison of parameters before and after the 3DPC exercise program (*n* = 37).

Parameter		Before	After	*p*
Cobb angle (°)	Thoracic	33.3 ± 12.4	28.3 ± 11.1	<0.01 *
	Lumbar	28.3 ± 11.2	24.1 ± 9.9	<0.01 *
Trunk rotation angle (°)	Thoracic	8.0 ± 4.3	4.7 ± 3.3	<0.01 *
	Lumbar	6.8 ± 4.3	3.2 ± 2.8	<0.01 *
Trunk expansion (cm)	Axilla	82.7 ± 6.1	83.2 ± 5.9	<0.01 *
	Xiphoid process	74.4 ± 6.5	75.2 ± 6.5	<0.01 *
	Waist	66.5 ± 6.0	68.3 ± 5.9	<0.01 *

All values (except *p*-values) are means ± standard deviations. * Significant at the 5% level.

## Data Availability

The data presented in this study are available on request from the corresponding author.

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
