# Peer review of "Effects of 3D Postural Correction and Abdominal Muscle Contraction on the Symmetry of the Transverse Abdominis and Spinal Alignment in Patients with Idiopathic Scoliosis"

_ijerph, 2023, doi:10.3390/ijerph20065016_

Round 1
Reviewer 1 Report
Despite the exciting theme and study limitations (without control group), this paper is good for publication and has need improvements;
The introduction is good and leaves the readers with the main point of the study.
The material and method need some corrections;
· The authors need to give more detail about Rigo classification. Also, the Rigo has a clinical classification type, which one was preferred by the authors?
· I think the information about table 1 and N value are wrong, please correct
· (Line 98-99 ) What is the specification of biofeedback unit? (mark, country, etc).
· (Line 120-133) Who did give or applied the exercises, and what is the clinician's experience?
· (Line 138-139) Please add a previous literature/article for reference point.
· (Line 159-160) The authors underestimated taking neutral inspiration, ı think the differences between neutral and max inspiration can show better trunk expansion.
· (Line 175-176) Please check the sentence. The t-test is not assessing the correlation and adds the name of the normality test (line 180).
· (Line 178-179) “A paired t-test to test whether there is a significant difference between the symmetry before and after your intervention” the font and grammar of this sentence need to correction.
· The authors did not present a sample size calculation. Please provide this information.
The results are clear and well-presented, need some minör changes;
· Please add min and max values, particularly for age and Cobb angle in Tables 2 and 3.
· What was the drop-out rate for the second experiment, The patients completed all exercise sessions?
The discussion and limitation need to expand with my suggestion.
Author Response
Response to Reviewer 1 Comments
Dear. Reviewer
The kind reply made my manuscript much better. Thanks to you, I studied a lot and was able to improve.
Thanks a lot
Regards,
Sungyoung
- The authors need to give more detail about Rigo classification. Also, the Rigo has a clinical classification type, which one was preferred by the authors?
Response: Thank for your review. As you suggested, I added about the more detail explanation. Rigo Classification is classified by clinical and radiological criteria. So, I preferred both criteria.
- I think the information about table 1 and N value are wrong, please correct
Response: Thank for your review. I’ve edited it correct.
- (Line 98-99) What is the specification of biofeedback unit? (mark, country, etc).
Response Line 98-99: Thank for your review. I've edited it as you suggested.
- (Line 120-133) Who did give or applied the exercises, and what is the clinician's experience?
Response Line 120-133: Thank for your review. I've edited it as you suggested.
- (Line 138-139) Please add a previous literature/article for reference point.
Response Line 138-139: Thank for your review. As you suggested, I added reference.
- (Line 159-160) The authors underestimated taking neutral inspiration, ı think the differences between neutral and max inspiration can show better trunk expansion
Response Line 159-160: Thank for your review. I understand the content that the reviewer is talking about very well. I will try to follow the direction you have suggested for future research.
- (Line 175-176) Please check the sentence. The t-test is not assessing the correlation and adds the name of the normality test (line 180).
Response Line 175-176: Thank for your review. I've edited it as you suggested.
- (Line 178-179) “A paired t-test to test whether there is a significant difference between the symmetry before and after your intervention” the font and grammar of this sentence need to correction.
Response Line 178-179: Thank for your review. I've edited it as you suggested.
- The authors did not present a sample size calculation. Please provide this information.
Response: Thank for your review. As you suggested, I added sample size calculation.
- Please add min and max values, particularly for age and Cobb angle in Tables 2 and 3.
Response: Thank for your review. I've edited it as you suggested.
- What was the drop-out rate for the second experiment, The patients completed all exercise sessions?
Response: All patients successfully completed all exercise sessions, despite a dropout rate of 20% having been used to calculate the necessary sample size.

Reviewer 2 Report
Thank you for the opportunity to review your manuscript, Effects of 3D postural correction and abdominal muscle con traction on the symmetry of the transverse abdominis and spi nal alignment in patients with idiopathic scoliosis.
The abstract needs to be better structured. It should be reworded.
If the sample included subjects aged 10 to 30 years, the authorisation of the legal guardian of the minor subjects should be specified. I understand this was done, but it is not stated in the manuscript.
Considering that the profile of the participants was the same, why was the ultrasound scan not performed on all participants and thus obtained a larger sample?
The location of the study is not specified.
Title of table 1. Is it correct?
Line 105. "To measure the TrA thickness, the perpendicular distance of the TrA at a site 1.5 cm". Based on which study?.
For the reliability of the measurements, it would be interesting to give the standard error of measurement (SEM) and the Minimum detectable difference (MDD).
Line 264-266 “ …exercise interventions are more beneficial from functional and psychological viewpoints, such as quality of life improvements” Why have these variables not been taken into account in the study?
Attaching some descriptive images of the exercises performed would improve the comprehension of the article.
Author Response
Response to Reviewer 2 Comments
Dear. Reviewer
The kind reply made my manuscript much better. Thanks to you, I studied a lot and was able to improve.
Thanks a lot
Regards,
Sungyoung
The abstract needs to be better structured. It should be reworded.
Response: Thank for your review. I've edited it as you suggested.
If the sample included subjects aged 10 to 30 years, the authorisation of the legal guardian of the minor subjects should be specified. I understand this was done, but it is not stated in the manuscript.
Response: Thank for your review. As you suggested, I added information about IRB
Considering that the profile of the participants was the same, why was the ultrasound scan not performed on all participants and thus obtained a larger sample?
Response: Thank for your review. The participants recruited for the primary experiment are different from those recruited for the following experiment.
The location of the study is not specified.
Response: Thank for your review. I've edited it as you suggested.
Title of table 1. Is it correct?
Response: Thank for your review. I’ve edited it correct.
Line 105. "To measure the TrA thickness, the perpendicular distance of the TrA at a site 1.5 cm". Based on which study?.
Response: Thank for your review. Additional studies that were based on were attached to the reference
For the reliability of the measurements, it would be interesting to give the standard error of measurement (SEM) and the Minimum detectable difference (MDD).
Response: Thank for your review. I've edited it as you suggested. In the study, the transverse abdominis symmetry ratio was marked, so data on thickness measurement was not mentioned and not marked.
Line 264-266 “ …exercise interventions are more beneficial from functional and psychological viewpoints, such as quality of life improvements” Why have these variables not been taken into account in the study?
Response: Thank for your review. Braces and specially designed physiotherapy exercises are commonly used as conservative treatments for scoliosis. CCs used in this study are considered to function as pads that provide pressure and force, similar to the principle of braces. While braces have their advantages, they also have disadvantages in terms of functional and psychological aspects. Therefore, the primary experiment aimed to demonstrate the clinical significance of 3DPC using CCs and AMC, which are as effective as braces. So the functional and psychological viewpoints were not included as variables in this study.
Attaching some descriptive images of the exercises performed would improve the comprehension of the article.
Response: Thank for your review.

Reviewer 3 Report
The manuscript represents the study with the aim to investigate the effect of 3D postural correction using corrective cushions and abdominal muscle contraction on thickness symmetry of the TrA muscle and spine alignment in idiopathic scoliosis patients. These are my comments and suggestions:
Abstract:
Please, indicate which abdominal muscle contraction in your aim of the study. Also, indicate - were the same patients were included in the first and second experiment? What was the design of the study? Include quantitative data, not only p-value.
Introduction:
Indicate, which abdominal muscles (line 42). I suggest to add information whether the muscles were thinner on both sides or only unilaterally (line 47).
Methods:
What was the design of the study? Were the same patients in the second experiment? How did you estimate your sample size? Did you seek ethical permission for this study?
Please, use reporting guidelines appropriate for your study design and rewrite your methods accordingly.
Results:
No comments.
Discussion and conclusion:
Please, compare your results with other studies. Clarify if your experiments involved same participants. Use reporting guidelines for this type of study design and rewrite relevant sections of the manuscript.
Author Response
Response to Reviewer 3 Comments
Dear. Reviewer
The kind reply made my manuscript much better. Thanks to you, I studied a lot and was able to improve.
Thanks a lot
Regards,
Sungyoung
Dear Authors,
Thanks for the revised manuscript, please see below for my further comments:
- The manuscript represents the study with the aim to investigate the effect of 3D postural correction using corrective cushions and abdominal muscle contraction on thickness symmetry of the TrA muscle and spine alignment in idiopathic scoliosis patients. These are my comments and suggestions:
Abstract:
Please, indicate which abdominal muscle contraction in your aim of the study. Also, indicate - were the same patients were included in the first and second experiment? What was the design of the study? Include quantitative data, not only p-value.
Response
Thank for your review.
a) Rather than a specific contraction of a particular abdominal muscle, it can be understood as an overall contraction of the abdominal muscles located in the lateral abdominal wall, using a pressure biofeedback unit, as described in design and setting.
b) The patients in the primary experiment and the following experiment are different.
c) Two-way analysis of variance (ANOVA) was performed to compare the symmetry of TrA thickness on the convex and concave sides of the lumbar curve during AMC/non-AMC without 3DPC and the TrA symmetry on the convex and concave sides of the lumbar curve during AMC/non-AMC with 3DPC using a CCs. In order to compare the differences in muscle thickness symmetry for AMC presence or absence and 3DPC presence or absence, a paired t-test was conducted separately for each group. A paired t-test was conducted to examine whether there was a significant difference in symmetry between the pre- and post-intervention measures.
d) The abstract should be a total of about 200 words maximum so I'm sorry I can't add more words.
Introduction:
Indicate, which abdominal muscles (line 42). I suggest to add information whether the muscles were thinner on both sides or only unilaterally (line 47).
Response line 42: Thank for your review. I've edited it as you suggested.
Response line 47: I've edited it as you suggested.
Methods:
What was the design of the study? Were the same patients in the second experiment? How did you estimate your sample size? Did you seek ethical permission for this study?
Please, use reporting guidelines appropriate for your study design and rewrite your methods accordingly.
Response
Thank for your review.
a) Two-way analysis of variance (ANOVA) was performed to compare the symmetry of TrA thickness on the convex and concave sides of the lumbar curve during AMC/non-AMC without 3DPC and the TrA symmetry on the convex and concave sides of the lumbar curve during AMC/non-AMC with 3DPC using a CCs. In order to compare the differences in muscle thickness symmetry for AMC presence or absence and 3DPC presence or absence, a paired t-test was conducted separately for each group. A paired t-test was conducted to examine whether there was a significant difference in symmetry between the pre- and post-intervention measures.
b) The patients in the primary experiment and the following experiment are different.
c) For the primary experiment, the sample size was calculated to compare the means be-tween conditions, with an effect size of 0.9, a significance level of 0.05, a power of 80%, and a dropout rate of 10%. For the following experiment, the sample size was calculated to compare the intervention effect, with an effect size of 0.9, a significance level of 0.05, a power of 80%, and a dropout rate of 20%.
d) This study was divided into the primary experiment and the following experiment. The primary experiment is an observational study and the following experiment is an experimental study design.
Results:
No comments.
Discussion and conclusion:
Please, compare your results with other studies. Clarify if your experiments involved same participants. Use reporting guidelines for this type of study design and rewrite relevant sections of the manuscript.
Response
Thank for your review.
a) I've edited it as you suggested.
b) I've edited it as you suggested in limitations.
c) This study was divided into the primary experiment and the following experiment. The primary experiment is an observational study and the following experiment is an experimental study design.

Round 2
Reviewer 2 Report
The authors have answered all my questions. They have also improved the manuscript enough to recommend it for publication.
Author Response
Response to Reviewer 2 Comments
Dear. Reviewer
The kind reply made my manuscript much better. Thanks to you, I studied a lot and was able to improve higher.
Thanks a lot
Regards,
Sungyoung
The authors have answered all my questions. They have also improved the manuscript enough to recommend it for publication.
Response: Thank for your review and reply.

Reviewer 3 Report
I am satisfied with the revised manuscript.
Author Response
Response to Reviewer 3 Comments
Dear. Reviewer
The kind reply made my manuscript much better. Thanks to you, I studied a lot and was able to improve higher.
Thanks a lot
Regards,
Sungyoung
I am satisfied with the revised manuscript.
Response: Thank for your review and reply.
